Increased chemical acetylation of peptides and proteins in rats after daily ingestion of diacetyl analyzed by Nano-LC-MS/MS

Jedlicka Leticia Dias Lima 1 2
Guterres Sheila Barreto 1 3
Balbino Aleksandro Martins 1
Neto Giuseppe Bruno 1
Landgraf Richardt Gama 1
Fernandes Liliam 1
Carrilho Emanuel 4
Bechara Etelvino José Henriques 1 5
Assuncao Nilson A. nilson.assuncao@gmail.com nilson.assuncao@unifesp.br 1
1 Institute of Environmental, Chemical and Pharmaceutical Sciences, Universidade Federal de São Paulo , Diadema , SP , Brazil
2 Institute of Studies in Health and Biological, Collective Health, Universidade Federal do Sul e Sudeste do Pará , Maraba , PA , Brazil
3 Department of Chemistry, Fundação Universidade Federal de Rondônia , Porto Velho , RO , Brazil
4 São Carlos Institute of Chemistry, Universidade de São Paulo , São Carlos , SP , Brazil
5 Department of Fundamental Chemistry, Institute of Chemistry, Universidade de São Paulo , São Paulo , SP , Brazil
Uversky Vladimir
Electronic publication date: 2018 Apr 25
Publication date: 2018
Volume: 6
Electronic Location ID: e4688
Received 2018 Mar 2; Accepted 2018 Apr 10
Copyright: ©2018 Jedlicka et al.
Copyright year: 2018
Copyright holder: Jedlicka et al.
License: This is an open access article distributed under the terms of the Creative Commons Attribution License, which permits unrestricted use, distribution, reproduction and adaptation in any medium and for any purpose provided that it is properly attributed. For attribution, the original author(s), title, publication source (PeerJ) and either DOI or URL of the article must be cited.
License URL: https://creativecommons.org/licenses/by/4.0/

Keywords: Radical acetylation, Diacetyl, Food additive, Lung diseases, Proteomics.

Funding: São Paulo Research Foundation (FAPESP) 2012/02514-9 2013/07763-0 2010/01404-0 Brazilian Innovation Agency (FINEP) This material is based upon work supported by the São Paulo Research Foundation (FAPESP, grants no. 2012/02514-9, 2013/07763-0, and 2010/01404-0) and the Brazilian Innovation Agency (FINEP). The funders had no role in study design, data collection and analysis, decision to publish, or preparation of the manuscript.

==============================
Background

Acetylation alters several protein properties including molecular weight, stability, enzymatic activity, protein–protein interactions, and other biological functions. Our previous findings demonstrating that diacetyl/peroxynitrite can acetylate L-lysine, L-histidine, and albumin in vitro led us to investigate whether diacetyl-treated rats suffer protein acetylation as well.

Methods

Wistar rats were administered diacetyl daily for four weeks, after which they were sacrificed, and their lung proteins were extracted to be analysed by Nano-LC-MS/MS (Q-TOF). A C18 reversed-phase column and gradient elution with formic acid/acetonitrile solutions from 2 to 50% over 150 min were used to separate the proteins. Protein detection was performed using a microTOF-Q II (QTOF) equipped with captive source and an electrospray-ionization source. The data from mass spectrometry were processed using a Compass 1.7 and analyzed using Protein Scape, software that uses Mascot algorithms to perform protein searches.

Results

A set of 3,162 acetylated peptides derived from 351 acetylated proteins in the diacetyl-treated group was identified. Among them, 23 targeted proteins were significantly more acetylated in the diacetyl-treated group than in the PBS control. Protein acetylation of the group treated with 540 mg/kg/day of diacetyl was corroborated by Western blotting analysis.

Conclusions

These data support our hypothesis that diacetyl exposure in animals may lead to the generation of acetyl radicals, compounds that attach to proteins, affecting their functions and triggering adverse health problems.

Introduction

Diacetyl is a flavoring commonly used in foodstuffs, as it lends a buttery flavor to products such as popcorn, coffee blends, cakes, cookies, wines and other goods (McCoy et al., 2017; Shibamoto, 2014; Ryan et al., 2014; Papetti, Mascherpa & Gazzani, 2014; Park, Gilbert & Whittaker, 2018; More, Raza & Vince, 2012). It is a volatile α-dicarbonyl and a highly electrophilic compound (Ryan et al., 2014) approved worldwide for use by food industries, despite ongoing health concerns dating back to 1986 (NIOSH, 1986), when the first cases of bronquiolitis obliterans involving diacetyl emerged (Park, Gilbert & Whittaker, 2018; Kreiss, 2014; Kanwal et al., 2011).

Recently, we reported that the reaction of peroxynitrite with α-dicarbonyls, namely diacetyl and methylglyoxal, in aerated phosphate buffer pH 7.4 results in the acetylation of free amino acids, peptides and proteins added to the reaction mixture (Alves et al., 2013; Massari et al., 2010; Massari et al., 2011; Tokikawa et al., 2014). This reaction is initiated by nucleophilic addition of peroxynitrite to the carbonyl group of the α-dicarbonyl compound yielding a peroxynitroso adduct, whose homolysis yields acetyl radicals. Dissolved molecular oxygen adds to the radical to ultimately produce acetate from diacetyl or acetate and formate from methylglyoxal (Massari et al., 2010; Massari et al., 2011). Formyl radical intermediate generated by methylglyoxal/peroxynitrite was shown to add to the α-amino group of L-lysine-containing synthesized tetrapeptides (Tokikawa et al., 2014). On the other hand, diacetyl/peroxynitrite-generated acetyl radicals have proven been proven to attack both the α- and ε-amino groups of free and blocked L-Lys, L-Lys-containing peptides and serum albumin (Alves et al., 2013). These findings have raised the hypothesis that radical acetylation of proteins contributes to transacetylase–promoted, post-translational protein modifications at sites where both methylglyoxal or diacetyl and peroxynitrite are present (Alves et al., 2013; Massari et al., 2010; Massari et al., 2011; Tokikawa et al., 2014). From these facts, the competition of chemical (induced by diacetyl) and enzymatic (occurring naturally in organisms) acetylation can be inferred, with the former process contributing to the increase of total protein acetylation.

Another source of acetylation in vivo is found through the action of acetyltransferases. These enzymes reversibly catalyze the transfer of the acetyl group from acetyl-CoA to the ε-amino group of protein lysine residues (Drazic et al., 2016), a process promoted by lysine acetyltransferase and lysine deacetylase (Meng et al., 2018; Arif, Selvi & Kundu, 2010; Iyer, Fairlie & Brown, 2012) at the N-terminus during the synthesis of proteins. Protein acetylation is highly conserved in eukaryotes and prokaryotes than phosphorylation, but it is less common than phosphorylation and ubiquitination (McEwan & Dikic, 2011). Acetylation can reportedly alter the protein function, size, enzymatic activity, stability, protein-protein interactions and other protein properties. When acetyltransferase is deregulated, and lysine acetylation is increased, modifications may occur in genes and in the regulatory machinery, resulting in the manifestation of tumours in cells (Drazic et al., 2016). On the other hand, ATP-dependent acetylation has recently been reported to play a role in many cellular processes such as catalytic activity, immune responses and metabolic processes, including the generation of precursors of “energy-rich” metabolites such as acetylphosphate (acP). AcP-dependent acetylation tends to govern the translation of nucleotides, purine and pyrimidine metabolism and degradation of RNA (Kuhn et al., 2014).

In this work, we use proteomic and western blotting techniques to investigate if diacetyl is also capable of leading to increases in vivo protein acetylation. Based on our findings, we support the hypothesis that diacetyl exposure in animals may lead to increases in protein acetylation, which may affect protein functions and trigger adverse health problems.

Methods

Animal treatment

All animals were fed ad libitum and kept in a cabinet at 50–70% humidity, at a temperature of 19–26 °C in a cycle of 12 h light/12 h dark. This study adheres to the guidelines established by the Brazilian College of Animal Experimentation (COBEA) and was approved by the Ethical Committee of the School Medicine of the Federal University of São Paulo (UNIFESP, protocol no. 1949-11).

Eight-to-twelve-week-old male Wistar rats (250–300 g) were divided into two groups (6 animals each). The control group received phosphate-buffered saline (PBS), while the treated group received 540 mg/kg/day of diacetyl (Cat B8530-7; Sigma Aldrich, St. Louis, MO, USA) dissolved in PBS Both groups were dosed using gavage. The concentration of 540 mg/Kg/day of 2,3-butanedione and the treatment period of four weeks was based on the experiment conducted by Colley and Cols (Colley et al., 1969).

After four weeks of treatment, the animals were anesthetized with ketamine and xylazine (Sigma Aldrich, St. Louis, MO, USA) and sacrificed. The lung tissue was collected and immediately frozen in liquid nitrogen and stored at –80 °C.

Preparation of lung extracts

Tissue preparation

Frozen lungs were ground into a fine powder in liquid nitrogen using a mortar and pestle. The homogenization process was used to avoid the activation of proteases and prevents protein degradation. The sample was lyophilized prior to analysis in order to remove residual water and stabilize the sample for handling at room temperature, thereby facilitating the weighing process and preparation of the sample.

Protein extraction optimization

Due to the wide range of proteins and interfering substances in the final extracts, the samples were obtained in the following three steps prior to proteomics analysis: tissue disaggregation and cell homogenization; protein extraction from the biological matrix; and protein precipitation and solubilisation in a urea buffer.

Three methods of protein extraction were tested to quantify the amount of protein in the lysates before and after precipitation (Table 1). Thirteen milligrams of lyophilized lung suspended in one mL of extraction buffer were used.

Table 1 Composition of the tested buffers.

Composition	Buffer 1a	Buffer 2b	Buffer 3c	
Buffering agent	Tris 25 mM	Tris 25 mM	Tris 25 mM	
Surfactant	CHAPS 2%	CHAPS4% TRITON 1%	CHAPS 4%	
Protease inhibitors	Aprotinine, leupeptine, pepstatine, benzamidine and PMSF	Aprotinine, leupeptine, pepstatine, benzamidine and PMSF	Aprotinine, leupeptine, pepstatine, benzamidine and PMSF	
Chaotropes			7 M Urea	
			2 M Thiourea	
Reducer			65 mM DTT	
Notes.

a Buffer 1 with power low solubility (few surfactants without chaotropes).

b Buffer 2 with median solubilizing power (more surfactants).

c Buffer 3 with maximum solubilization power (presence of chaotropes, surfactants and reducers).

The lung powder was suspended and shaken for 1 h at 4 °C. After centrifugation (10 min, 5,000 × RPM, 4 °C), 200 µL of the supernatant was mixed with 800 µL of DTT solution in cold acetone (2 mg/mL) and incubated overnight at −20 °C. Afterwards, the samples were centrifuged (10 min, 16,000 × RPM, 4 °C), and the sediments were washed four times with the DTT solution, dried in vacuum and solubilized in urea buffer (7 mol L−1 urea, 2 mol L−1 thiourea, 4% CHAPS). The total protein concentration was determined by the Bradford method (Bradford, 1976).

Tryptic digestion

Prior to tryptic digestion, polypropylene microtubes were individually filled with an extract aliquot containing 250 µg of the lung protein. The samples were reduced with Dithiothreitol (DTT) solution until achieving a final concentration of 5 mmol L−1, and they were then incubated for 25 min at 56 °C. To achieve alkylation in the samples, iodoacetoamide (IAA) was added until reaching a final concentration of 14 mmol L−1. The samples were then incubated for 30 min at room temperature and protected from light. Afterward, they were diluted until the concentration of urea was reduced to 1,600 mmol L−1, and a CaCl2 solution was added until reaching a final concentration of 1 mmol L−1 of CaCl2.

The enzymes trypsin and LysC endoproinase were added in the ratio of 1:50 of (enzyme: substrate). The samples were incubated for 18 h at 37 °C. The enzyme reaction was stopped by adding TFA (trifluoroacetic acid) at the final concentration of 0.4%. The samples were centrifuged at 2,500 rpm for 10 min at room temperature, and the pellet was discarded. The sample was evaporated until the volume was reduced to approximately 50 µl using a vacuum concentrator (Speed Vacuum; Thermo Fisher Scientific, Waltham, MA, USA). Finally, 50 µl of 0.5% trifluoroacetic acid (TFA) were added thereto. Detergents were removed from the sample using a Pierce detergent removal spin column (# 87776; Pierce Biotechnology/Thermo Fisher Scientific, Waltham, MA, USA), which was used according to the manufacturer’s specifications; the samples were filtered through a 22 µm PVDF syringe filter stocked at 4 °C for mass spectrometry analysis.

NanoLC–ESI/MS/MS analyses

In this study, on average six biological replicates and two replicate techniques were used. However, due to technical problems, we used 11 replicates of the control group and 10 replicates of the group treated with 540 mg/kg/day of diacetyl. Each trypsinized sample was dissolved with 100 µL of a mixture of water/acetonitrile/TFA (949:50:1 v/v). All analyses were performed using a Nano-UHPLC Advance (Bruker Daltonics, Bremen, Germany) equipped with a pump, an auto sampler, and a thermostatically controlled column compartment. A C18 reversed-phase column (Magic C18 AQ, P/N: CP3/61271/00; Michrom, Boise, ID, USA), particle size 3 µm, internal diameter 0.1 mm, length 100 mm was used. The column temperature was kept at 40 °C. Samples were separated using a gradient mobile phase consisting of (A) formic acid/ACN/H2O (1:20:979) and (B) formic acid/ACN/H2O (1:950:50) in a gradient elution from 2 to 50 % over 150 min, as a graph in SM1. The flow rate was set at 0.500 µL/min, and the injection volume was 5 µL. Detections were performed using a micrOTOF-Q II (Bruker Daltonics, Billerica, MA, USA), an accurate mass instrument equipped with captive source (Bruker Daltonics, Billerica, MA, USA) and an electrospray-ionization source (ESI). The mass spectrometer was running in positive mode, with the desolvation temperature at 180 °C and the nebulizer set at 500 V and 0.4 bars. All the operations, acquisition, and analysis of data were controlled by Hystar software version 1.7 (Bruker Daltonics, Billerica, MA, USA). For MS/MS analyses, five precursor ions were automatically selected to undergo collision and fragmentation with argon gas (≥ 2 L/min). Mass spectra were collected between 50 to 3,000 m/z, and calibration was performed at the beginning of every day using the Tune-Mix ESI-G (Agilent Technologies, Santa Clara, CA, USA). The collision energy was 12 eV, collision RF 600 Vpp, transfer time 140 µs, and pre-pulse storage 14 µs. MS/MS parameters were three precursor ions, absolute threshold 2,000 cts, smart exclusion 5 ×, excluded after three spectra, and released after 1 min. The tune parameters were Funnel 1RF 300 Vpp, Funnel 2 RF 400 Vpp, hexapole RF 400 Vpp, quadrupole ion energy 6.0 eV, and low mass 300 m/z. The TOF (time of flight) conditions included the following: repetition rate 5 kHz, sample rate 2 Ghz, flight tube 8,600 V, reflector 1,700 V, detector source 1,700 V, and detector TOF 2140 V. Argon was used as a collision gas at a pressure of 2 ×10−6 mbar, and the collision energy values were 10–200 eV.

Bioinformatic analysis

Data deconvolution and database search

Data from mass spectrometry were processed using a Compass 1.7 for OTOF (Bruker Daltonics, Billerica, MA, USA) and deconvoluted to generate a file compatible with Mascot. This file was analyzed using Protein Scape (Bruker Daltonics, Billerica, MA, USA), a program that uses Mascot algorithms to perform the search. The database used was Swissprot, an annotated protein sequence database. The taxonomy was rattus, and the enzyme was trypsin with two missed cleavages. The fixed modification was carbamidomethylation, and the variable modifications were oxidation of methionine and acetylation of lysine and arginine. Mass tolerance modification was 150 ppm to 1 Da. Mascot analysis of all proteins (p < 0.05) used a minimum score of 35.

Protein network analysis

The protein-protein interaction analysis was performed using Cytoscape 3.3.0 software (http://www.cytoscape.org/) (Shannon et al., 2003), and the protein interaction network was obtained from the STRING 8.2 database (http://string-db.org/) (Szklarczyk et al., 2011). STRING 8.2 uses the metric of “confidence score” to define the confidence of the interactions. We selected only the interactions with proteins identified in our analyses.

Orthologs analysis

Orthologs were subjected to Gene Ontology (GO) term analysis based on PANTHER classification online tools (http://pantherdb.org/). To determine the biochemical functions of acetylated proteins detected in the lungs of the group treated with diacetyl, GO was performed using IDs with the Rattus norvegicus genome found in the Uniprot database. This particular database was chosen as the reference database for the output report of biologicals process, proteins class, cellular components, pathways and molecular functions (Mi et al., 2013). These analyses were performed to acquire insights of the acetylation involved in the functions and pathways of proteins.

Analysis of sequence model around acetylated lysine

The software motif-x was employed to determine specific sequences of amino acid (15 amino acids upstream and downstream of the acetylation site) in all protein sequences acquired from NanoLC-MS/MS analysis. The entire database (IPI Rat Proteome) was used as a background database parameter, and the significance was 0.000001 (Chou & Schwartz, 2011; Schwartz & Gygi, 2005).

Western blotting

Samples containing 25 µg of proteins from lung homogenate were subjected to 12% SDS–PAGE electrophoresis and electroblotted onto a nitrocellulose membrane (Millipore, USA). Following the blocking and washing steps, the membranes were incubated with the primary Acetylated-Lysine antibody (Cell Signalling, Danvers, MA, USA) and anti-rabbit IgG HRP-linked (Cell Signalling, Danvers, MA, USA) as a secondary antibody. The membranes were then detected using a chemiluminescence kit “Pierce ECL Plus Western Blotting Substrate” (Thermo Scientific, Waltham, MA, USA) and chemiluminescence software (GeneGnome System/Gene Tools Software; Syngene, Cambridge, UK).

Results

Protein extraction optimization

We performed three different protein extraction methods to determine which one yielded the most consistent results when reproduced. That method was then adopted, allowing for increased accuracy in the estimation of protein amounts from the lysate extracts. Figure 1 shows a comparison of the amount of proteins between lung lysate and the solution of precipitate lung proteins resuspended.

Figure 1 Comparison of the amount of total protein present in the lysate and the resolubilizated proteins (precipitate proteins resuspended) using lung tissue.

MS/MS analyses of rat lung proteins

In this work, qualitative proteomic analysis was used, specifically the bottom up technique. The lung extracts were analyzed by NanoLC-MS/MS, and significant differences were shown among the protein profiles in the control and diacetyl-treated groups. Acetylation was set in the search engine as a variable modification, and overall, the analyses showed 10,302 peptides identified as belonging to 603 proteins in lung tissue. A set of 327 acetylated proteins in the control group and 351 proteins in the group treated with 540 mg/kg/day of diacetyl were detected. This increase in acetylation can occur either enzymatically or chemically, a phenomenon presently demonstrated in this work.

In this experiment, we identified 93 proteins which were common between the control group and the 2,3-butanedione treated group. After this identification, we verified the peptides present in these 93 proteins, the peptides common among the groups and the incidence of acetylation in these peptides. This peptide analysis was performed to ensure that the acetylations found were due to the ingestion of 2,3-butanedione. We only validated the acetylations that were exclusively present in the treated group or that were expressed in a larger number in the treated group compared to the control group. After this analysis of the acetylated peptides, we selected 23 proteins, to which these acetylated peptides belong, and named them as ‘target proteins’.

Analysis of proteins and peptides revealed that acetylation is more abundant in the group treated with 2,3-butanedione than in the control group. The proteins that exhibited this pattern of acetylation, described earlier as ‘target proteins’, and their respective peptides are described in Table 2, which provides target protein identification and their respective peptide scores in both groups as well as descriptions of the peptide acetylation positions.

Table 2 Target proteins, acetylated proteins in group treated with diacetyl but non-acetylated in control group.

Protein I.D.	Gene I.D.	Meta score control	Meta score treated	Peptides control	Peptides treated	*SC [%] control	*SC [%] treated	
AL1A1_RAT	AL1A1	269.1	253.5	11	9	37.1	31.7	
ANXA2_RAT	ANXA2	539.6	284.6	13	13	44.5	43.7	
ANXA5_RAT	ANXA5	411	265	10	10	37.3	43.6	
ASSY_RAT	ASS1	71	1,030.1	3	21	20.6	64.3	
BHMT1_RAT	BHMT1	172.3	834.8	7	24	30	63.1	
CALR_RAT	CARL	277.1	607.2	8	13	31.7	49.5	
CES1D_RAT	CES1D	598.6	571.1	15	16	38.8	50.8	
DYH1_RAT	DNAHC1	1,391.9	1,603.7	77	89	23.5	27.7	
EF2K_RAT	EEF2K	354	304.5	18	17	38.5	34.1	
ENPL_RAT	HSP90B1	742.6	492.2	25	16	29.6	23.5	
EPHA6_RAT	EPHA6	486.4	361.4	27	20	32.4	24.5	
FABPL_RAT	FABPL	120.5	779.4	3	16	36.2	81.9	
G3P_RAT	GAPDH	486.7	763	14	18	53.8	62.2	
OGA_RAT	MEGEA5	308.5	353.6	16	19	26.7	31.6	
PARK7_RAT	PARK7	173.3	187.5	4	10	28.6	51.3	
PRC2A_RAT	PRRC2A	776.2	670.4	43	36	25.3	23.6	
SI1L1_RAT	SI1L1	575.9	672.1	32	37	27.9	30.7	
STIP1_RAT	STIP1	189.5	258.9	10	14	22.1	30	
SYPM_RAT	PARS2	171.2	183.4	9	10	32.6	46.3	
TBB4B_RAT	TUBB4B	801.7	463.1	20	15	52.1	48.8	
TPP1_RAT	TPP1	99.3	109.3	5	6	9.9	11.7	
UBR4_RAT	UBR4	1,600.3	1,813.1	90	103	24.2	30.2	
UD2B2_RAT	UGT2B	390.2	705.3	21	25	44.7	59.1	

Some peptides showed post-translational modifications, and these peptides are listed in Table 3. As expected, L-lysine appears to be the predominant acetylated amino acid in the peptide sequence, although arginine and histidine residues were found to be acetylated as well.

Table 3 Peptides from acetylated proteins in group treated with diacetyl but non-acetylated in control group.

Protein I.D.	Gene I.D.	Peptide sequence	Peptide meta score control	Peptide meta score treated	Acetylation treated group	
AL1A1_RAT	ALDH1A1	-.MSSPAQPAVPAPLANLKIQHTK.I	15.4	16.1	7; 20; 22	
ANXA2_RAT	ANXA2	K.ELPSAMKSALSGHLETVMLGLLK.T	15	21.7	3; 23	
ANXA2_RAT	ANXA2	K.ELPSAMKSALSGHLETVMLGLLK.T	15	16.6	3; 23	
ANXA2_RAT	ANXA2	K.GVDEVTIVNILTNR.S	71.9	18.2	14	
ANXA2_RAT	ANXA2	K.SALSGHLETVMLGLLK.T	94	18.6	6	
ANXA5_RAT	ANXA5	K.YMTISGFQIEETIDRETSGNLENLLLAVVK.S	16.4	16.7	15	
ANXA5_RAT	ANXA5	K.YMTISGFQIEETIDRETSGNLENLLLAVVK.S	16.4	17.6	5; 30	
ASSY_RAT	ASS1	R.GIYETPAGTILYHAHLDIEAFTMDR.E	39.8	16.1	13; 5	
BHMT1_RAT	BHMT	R.IASGRPYNPSMSKPDAWGVTK.G	16.3	17.5	5	
BHMT1_RAT	BHMT	R.IASGRPYNPSMSKPDAWGVTK.G	15.4	15.4	21; 30	
CALR_RAT	CALR	K.HEQNIDCGGGYVK.L	33	85.7	13	
CES1D_RAT	CES1D	K.GKVLGK.Y	24.4	15.1	2	
CES1D_RAT	CES1D	R.SHRDAGAPTFMYEFEYRPSFVSAMRPK.T	18.5	22.7	2; 25	
CES1D_RAT	CES1D	R.SHRDAGAPTFMYEFEYRPSFVSAMRPK.T	18.5	15.7	2; 7; 25	
DYH1_RAT	DNAH1	R.SSLTRLASHMAEYECFQVELSK.N	19	16.7	5	
EF2K_RAT	EEF2K	R.SGDLYTQAAEAAMEAMK.G	30.7	21.1	7	
ENPL_RAT	HSP90B1	R.MMKLIINSLYK.N	18.8	16.1	1; 3	
EPHA6_RAT	EPHA6	R.EASIMGQFDHPNIIRLEGVVTK.R	18.3	16.8	0; 5	
EPHA6_RAT	EPHA6	K.SVTEFNGDTITNTMTLGDIVYK.R	28.2	50.8	22	
FABPL_RAT	FABPL	K.SVTEFNGDTITNTMTLGDIVYK.R	16	36.2	22	
FABPL_RAT	FABPL	K.YQVQSQENFEPFMK.A	28.2	33.9	4	
G3P_RAT	GAPDH	K.RVIISAPSADAPMFVMGVNHEK.Y	18.6	23.1	1; 20; 22	
G3P_RAT	GAPDH	K.RVIISAPSADAPMFVMGVNHEK.Y	18.6	15.8	20; 22	
G3P_RAT	GAPDH	K.RVIISAPSADAPMFVMGVNHEK.Y	18.6	15.8	20; 22	
G3P_RAT	GAPDH	K.RVIISAPSADAPMFVMGVNHEK.Y	18.6	23.1	1; 20; 22	
OGA_RAT	MGEA5	K.LDQVSQFGCRSFALLFDDIDHNMCAADK.E	20	15.4	21; 28	
PARK7_RAT	PARK7	K.GAEEMETVIPVDIMR	28.2	16.1	5; 6	
PRC2A_RAT	PRRC2A	K.ALYPGALGRPPPMPPMNFDPRWMMIPPYVDPR.L	17.9	30.3	9	
PRC2A_RAT	PRRC2A	K.ALYPGALGRPPPMPPMNFDPRWMMIPPYVDPR.L	17.9	18.9	9	
PRC2A_RAT	PRRC2A	K.ALYPGALGRPPPMPPMNFDPRWMMIPPYVDPR.L	17.9	16.6	21	
PRC2A_RAT	PRRC2A	K.ALYPGALGRPPPMPPMNFDPRWMMIPPYVDPR.L	17.9	26.6	21; 32	
PRC2A_RAT	PRRC2A	K.ALYPGALGRPPPMPPMNFDPRWMMIPPYVDPR.L	17.9	19.4	32	
PRC2A_RAT	PRRC2A	K.ALYPGALGRPPPMPPMNFDPRWMMIPPYVDPR.L	17.9	15.1	9	
PRC2A_RAT	PRRC2A	K.AVGTPGGNSGGAGPGISTMSRGDLSQR.A	18.4	22	21; 27	
PRC2A_RAT	PRRC2A	R.ERSDSGGSSSEPFER.H	17.1	15.4	15	
SI1L1_RAT	SIPA1L1	K.EKSKPYPGAELSSMGAIVWAVR.A	15.6	19.4	2	
SI1L1_RAT	SIPA1L1	K.SLPLRRPSYTLGMK.S	19.5	16.8	5	
STIP1_RAT	STIP1	R.RAMADPEVQQIMSDPAMR.L	20.7	18.9	1; 8	
STIP1_RAT	STIP1	R.RAMADPEVQQIMSDPAMR.L	20.7	20.4	8	
SYPM_RAT	PARS2	K.GIEVGHTFYLGTKYSSIFNAHFTNA HGESLLAEMGCYGLGVTR.I	17.8	15	21; 26	
TBB4B_RAT	TUBB4B	R.INVYYNEATGGKYVPR.A	21.9	15.4	6; 12	
TPP1_RAT	TPP1	R.EREPELAQLLVDQIYENAMIAAGLVDDPR.A	15.2	19.3	29	
TPP1_RAT	TPP1	R.INTLQAIWMMDPK.D	15.9	15.1	3	
UBR4_RAT	UBR4	K.ALGTLGMTTNEKGQVVTK.T	15.7	21.7	2	
UBR4_RAT	UBR4	K.EKAAPPPPPPPPPLESSPR.V	18.3	18.1	2; 9	
UBR4_RAT	UBR4	K.EKEGESSGSQEDQLCTALVNQLNR.F	17.1	16.7	2; 24	
UBR4_RAT	UBR4	K.FLSRPALPFILRLLR.G	15.1	30	5; 12	
UBR4_RAT	UBR4	R.DNPEATQQMNDLIIGKVSTALK.G	28.8	17.2	6; 22	
UBR4_RAT	UBR4	R.DNPEATQQMNDLIIGKVSTALK.G	17.3	21	22	
UBR4_RAT	UBR4	R.MAGVMAQCGGLQCMLNRLAGVK.D	19.3	23.9	7	
UBR4_RAT	UBR4	R.TGSTSSKEEDYESDAATIVQK.C	19.4	17.3	7; 21	
UD2B2_RAT	UGT2B	K.EWDTFYSEILGRPTTVDETMSKVEIWLIR.S	15.2	16.8	12; 22	

The acetylation ratio from target proteins ratio was calculated in order to more effectively visualize the increase in acetylation. The increase in acetylation can be clearly seen in Table 4, which shows the increase in the acetylation ratio in the peptides identified in both groups. The student’s t-test was applied, and the difference was significant with p < 0.0001, demonstrating that there was a significant increase of the acetylation in these peptides.

Table 4 Ratio of acetylation in both groups: control and treated with 2,3-butanedione.

Protein I.D.	Gene I.D.	Peptide sequence	Acetylation ratio control group	Acetylation ratio treated group	
AL1A1_RAT	ALDH1A1	-.MSSPAQPAVPAPLANLKIQHTK.I	1	3	
ANXA2_RAT	ANXA2	K.ELPSAMKSALSGHLETVMLGLLK.T	0.5	0.666666667	
ANXA2_RAT	ANXA2	K.GVDEVTIVNILTNR.S	0	1	
ANXA2_RAT	ANXA2	K.SALSGHLETVMLGLLK.T	0	1	
ANXA5_RAT	ANXA5	K.YMTISGFQIEETIDRETSGNLENLLLAVVK.S	0	1	
ASSY_RAT	ASS1	R.GIYETPAGTILYHAHLDIEAFTMDR.E	0	2	
BHMT1_RAT	BHMT	R.IASGRPYNPSMSKPDAWGVTK.G	0	2	
CALR_RAT	CALR	K.HEQNIDCGGGYVK.L	0	1	
CES1D_RAT	CES1D	K.GKVLGK.Y	0	1	
CES1D_RAT	CES1D	R.SHRDAGAPTFMYEFEYRPSFVSAMRPK.T	2	1.5	
DYH1_RAT	DNAH1	R.SSLTRLASHMAEYECFQVELSK.N	0	1	
EF2K_RAT	EEF2K	R.SGDLYTQAAEAAMEAMK.G	0	1	
ENPL_RAT	HSP90B1	R.MMKLIINSLYK.N	0	2	
EPHA6_RAT	EPHA6	R.EASIMGQFDHPNIIRLEGVVTK.R	0	2	
FABPL_RAT	FABPL	K.SVTEFNGDTITNTMTLGDIVYK.R	0	1	
FABPL_RAT	FABPL	K.YQVQSQENFEPFMK.A	0	1	
G3P_RAT	GAPDH	K.RVIISAPSADAPMFVMGVNHEK.Y	1	1.5	
OGA_RAT	MGEA5	K.LDQVSQFGCRSFALLFDDIDHNMCAADK.E	0	2	
PARK7_RAT	PARK7	K.GAEEMETVIPVDIMR	1	0.5	
PRC2A_RAT	PRRC2A	K.ALYPGALGRPPPMPPMNFDPRWMMIPPYVDPR.L	1	0.5	
PRC2A_RAT	PRRC2A	K.AVGTPGGNSGGAGPGISTMSRGDLSQR.A	0	2	
PRC2A_RAT	PRRC2A	R.ERSDSGGSSSEPFER.H	0	1	
SI1L1_RAT	SIPA1L1	K.EKSKPYPGAELSSMGAIVWAVR.A	0	1	
SI1L1_RAT	SIPA1L1	K.SLPLRRPSYTLGMK.S	0	1	
STIP1_RAT	STIP1	R.RAMADPEVQQIMSDPAMR.L	0	1	
SYPM_RAT	PARS2	K.GIEVGHTFYLGTKYSSIFNAHFTNAH GESLLAEMGCYGLGVTR.I	1	2	
TBB4B_RAT	TUBB4B	R.INVYYNEATGGKYVPR.A	0	2	
TPP1_RAT	TPP1	R.EREPELAQLLVDQIYENAMIAAGLVDDPR.A	0	1	
TPP1_RAT	TPP1	R.INTLQAIWMMDPK.D	0	1	
UBR4_RAT	UBR4	K.ALGTLGMTTNEKGQVVTK.T	0	1	
UBR4_RAT	UBR4	K.EKAAPPPPPPPPPLESSPR.V	0	2	
UBR4_RAT	UBR4	K.EKEGESSGSQEDQLCTALVNQLNR.F	1	1	
UBR4_RAT	UBR4	K.FLSRPALPFILRLLR.G	0	2	
UBR4_RAT	UBR4	R.DNPEATQQMNDLIIGKVSTALK.G	2	3	
UBR4_RAT	UBR4	R.MAGVMAQCGGLQCMLNRLAGVK.D	0	1	
UBR4_RAT	UBR4	R.TGSTSSKEEDYESDAATIVQK.C	0	2	
UD2B2_RAT	UGT2B	K.EWDTFYSEILGRPTTVDETMSKVEIWLIR.S	0	2	
Mean	0.283783784 ± 0.092 (27)a	1.423423423 ± 0.104 (27)a	
Notes.

a Mean ± Std. Error(N).

Analysis of the distribution of acetylated proteins within the subcellular localization revealed that they were predominantly located in the cellular membrane and cytoplasm (53%). Nineteen percent are known to be present in the nucleus and 12% in the cytoskeleton, while 14% are in different organelles, including mitochondria and endoplasmatic reticulum.

Protein interaction analysis

Figure 2 consists of the acetylated protein network from treated group. This network represents this protein interaction. Nodes represent the proteins in the network, and each color represents a different situation in relation to protein acetylation, while the edges represent the interactions between the proteins.

Figure 2 Network of acetylated proteins in group treated with Diacetyl.

The green nodes are the proteins acetylated in group treated with Diacetyl but lack acetylation on treated groups. The red nodes are the proteins presents only in group treated with Diacetyl; the blue nodes are the proteins presents only in control group and the purple nodes are proteins present in both groups.

Orthology analyses

In order to reveal the involved cellular and metabolic processes as well as the subcellular location of the differentially expressed proteins in acetylation level with 2,3-butanedione treatment, the GO-based analysis was conducted.

Analysis of the Molecular Function (Fig. 3A) revealed catalytic activity (57%), followed by specific binding function (19%). The analyses of biological functions (Fig. 3B) indicated some processes in which acetylated proteins are involved, including cellular processes (28.6%) and responses to stimulus (14.3%). The top three protein classes (Fig. 3C) display hydrolase (19%), chaperone (14.3 %) and oxidoreductase (14.3 %) activities. The cellular component analyses (Fig. 3D) demonstrated that acetylated proteins belong to macromolecular complexes (9.5%), cell organelles (9.5%), extracellular region (4.8%) and other cell parts (19%).

Figure 3 Orthology analyses from more acetylated proteins in group treated with Diacetil than in control group.

(A) Molecular function; (B) biological process; (C) protein class; (D) cellular component.

Motif analysis of proteins containing arginine-, lysine- and histidine-acetylated peptides

In order to characterize the possible specific sequence motifs surrounding acetylated arginine, lysine and histidine residues in peptides of lung samples, a logo sequence to compute the likelihood of amino acids at the positions surrounding the acetylation site was generated. Ten significantly enriched motifs were obtained from all the identified acetylated sites including *K, *R, *H (*K represents the acetylated lysine, *R represents the acetylated arginine and *H represents the acetylated histidine). As shown in Fig. 4, logos with the highest scores were used and all motif analyses are available in SM 2–7. Figures 4A and 4D show the motif surrounding acetylated arginine in samples from the control and treated groups, respectively, and Figs. 4B and 4E show the motif surrounding acetylated lysine. A number of reports have already demonstrated the occurrence of acetylation in arginine residue (Mathews et al., 2010). Figures 4C and 4F portray the motif surrounding acetylated histidine from control and groups treated with diacetyl, respectively.

Figure 4 Motif analysis surrounding arginine, lysine and histidine acetylated peptides.

(A) Motif analysis of control group, surrounding acetylated arginine. (B) Motif analysis of control group, surrounding acetylated lysine. (C) Motif analysis of control group, surrounding acetylated histidine. (D) Motif analysis of group treated with diacetyl, surrounding arginine. (E) Motif analysis of group treated with diacetyl, surrounding lysine. (F) Motif analysis of group treated with diacetyl, surrounding histidine.

Western blotting

Western blotting experiments indicated that the acetylation level was significantly higher in the treated group as compared to the control group (Fig. 5). Figure 5A shows an increase in acetylation in bands that correspond between 35–70 KDa in lanes 5,7 and 8 that were filled with samples from the group treated with 2,3-butanedione. The wells filled with samples from the control group (lanes: 2–4) did not display the acetylation band.

Figure 5 Western blotting for acetylated proteins from lung samples.

(A) Western blotting 4 image: Lane 1: molecular weight; lanes 2–4: samples of the control group; lanes 5, 7, and 8: samples 5 of the diacetyl-treated group; and lanes 6, 9, and 10: sample buffer. (B) Western blotting quantification.

Statistical analysis by the Student t-test revealed that mean values of protein intensities and variances are significantly different, with p = 0.0091 for means and p = 0.0015 for variance. This Western blotting experiment data confirms the result of LC-MS/MS analysis, which revealed increases in protein acetylation from the group treated with 2,3-butanedione in comparison with the control group.

Discussion

Protein extraction optimization

The amount of proteins obtained from precipitated and resolubilizated proteins using three buffers. Buffers 1 and 2 were slightly more efficient than Buffer 3, which led us to choose Buffer 2 in all experiments.

MS/MS analyses of rat lung proteins

The results from provide evidence of an increase in protein acetylation in the group treated with diacetyl. Acetylation reportedly alters protein function, size, enzymatic activity, stability, protein-protein interactions and other protein properties. Some proteins regulate acetyltransferases and histone deacetylases and may induce acetylation of other proteins (Drazic et al., 2016). When acetyltransferases are deregulated, and lysine acetylation is increased, modifications may occur in genes and the regulatory machinery (Drazic et al., 2016). These data show that diacetyl- triggered protein acetylation takes place in different cell compartments and that it may be implicated in many cell functions.

Protein interaction analysis

The protein interaction analysis showed that some acetylated proteins are interconnected and/or connected with other proteins. To exemplify this interaction, we can cite the protein ASS1, found to be acetylated in the treated group, which interacts with both ALS and OTC. Present in our control group is the ALS enzyme, whose activity is regulated by acetylation, according to the UniProt database (http://www.uniprot.org). ASS1 interacts with OTC, which is present only in the diacetyl-treated group. OTC, one of the enzymes of the urea cycle, acts by detoxifying the excess of ammonium produced from amino acid catabolism and is negatively regulated by lysine acetylation (Yu et al., 2009).

Some acetylated proteins present in the network are involved in the cell redox balance (Drazic et al., 2016), in protein biosynthesis and have ATP and nucleotide binding activity, maturation, structural maintenance and regulation of specific proteins (Haase & Fitze, 2016), along with cellular processes such as the basal metabolism, immunogenicity, cell cycle progression, DNA repair and apoptosis (Michalak et al., 2009). Some proteins also induce anti-tumor immunity by inhibiting angiogenesis and have antioxidant activity in neurons and the heart, protecting against cell death (Kolisek et al., 2015; Singh et al., 2015). Additionally, the proteins play a cytoprotective role being a redox-responsive protein (Eltoweissy et al., 2011).

The increase in chemical acetylation of lung proteins of diacetyl-treated rats described here may be connected with the fact that diacetyl has been shown in vitro to generate acetyl radicals upon reaction with peroxynitrite, and more slowly with hydrogen peroxide (Tokikawa et al., 2014). The diacetyl/peroxynitrite system was then reported to promote acetylation of isolated amino acids, peptides and albumin. These data led us to postulate that post-translational chemical acetylation of proteins may contribute to enzymatic acetylation at sites where both diacetyl and peroxynitrite at inflammation are formed.

Orthology analyses

The Gene Ontology (GO) function analysis of the target proteins revealed the distribution and function of these proteins. Protein acetylation regulates enzyme activities that mediate, for instance, the degradation of proteasomes and lysosomes by neutralizing the lysine residues in the active sites, thereby causing conformational changes. In addition to regulating the catalytic activity of metabolic enzymes, acetylation controls substrate accessibility, blocks substrate binding to the enzyme and modulates enzyme subcellular localization (Xiong & Guan, 2012).

The most crucial pathways are those related to the oxidative stress response (P00046), which causes cellular damage. In a normal functioning cell, several transcription factors respond to oxidative stress by modulating the expression of genes whose products relieve the altered redox status.

Motif analysis of proteins containing arginine-, lysine- and histidine-acetylated peptides

The possible motifs surround acetylated arginine, lysine and histidine. Despite lysine being the more common site of protein acetylation, some studies have demonstrated that arginine can be acetylated as well, triggering biological responses (Rabbani et al., 2011; Slade, Subramanian & Thompson, 2014). Acetylation in both lysine and histidine residue was previously demonstrated in vitro (Alves et al., 2013; Massari et al., 2010; Massari et al., 2011; Tokikawa et al., 2014), which reinforces our results about acetylation in these residues.

Western blotting

Western blotting was used to confirm the increase of acetylation previously found by NanoLc-ms/ms experiments. We used a specific acetylation antibody to detect bands with a substantial increase in intensity in samples from the group treated with diacetyl that reveals the protein acetylation increase. These results confirm the protein acetylation identified by NanoLc-ms/ms analyses.

Conclusions

Altogether, our data strongly suggest that diacetyl gavage administered to rats may constitute a source of acetyl radical that can attack and acetylate lung proteins. It is tempting to hypothesize that this is a contribution mechanism for the reported toxicity of diacetyl in workers dealing with ‘buttered’ food who subsequently acquire bronquiolitis obliterans.

Herein, we first optimized extraction conditions for the lung proteins of Wistar rats, for both rats in the control group and those treated with diacetyl. Mass spectrometry results, confirmed by Western blotting analyses, revealed increased acetylation in the lung tissues of groups treated with 2,3-butanedione.

The proteins acetylated to different extents in the diacetyl-treated group were then related to reported interactions with other key proteins and enzymes of cell homeostasis. Diacetyl treatment, apparently, modifies the lung protein profile. Twenty-three diverse classes of proteins were found to undergo preferential acetylation. They are present in different regions of the cell and are involved in different molecular and biological processes. Our data indicate that the observed increased radical acetylation by diacetyl occurs randomly.

In a comprehensive view, we found more peptides acetylated in the group treated with diacetyl than in the control group. The expected acetylation of lysine residues also occurred in arginine and histidine, suggesting that unlike acetylase-driven acetylation of proteins, radical acetylation occurs randomly, modifying residues of both the N-terminal, the C-terminal and the side chain of basic amino acid residues. The Western blotting analysis clearly demonstrated increased protein acetylation due to the daily intake of diacetyl.

Our study is consistent with early in vitro studies that showed increases in protein acetylation in the presence of 2,3-butanedione. The data reported here reinforce our hypothesis that diacetyl exposure is capable of increasing protein acetylation in vivo, thus raising a potential for diacetyl, a highly electrophilic α-dicarbonyl industrial xenobiotic, to play a role in inflammatory bronquiolitis obliterans (Cavalcanti et al., 2012; Cummings et al., 2014; Egilman & Schilling, 2012).

Supplemental Information

Supplemental Information 1 NanoHPLC gradient elution

NanoHPLC gradient elution, gradient mobile phase consisting of (A) formic acid/ACN/ H2O (1:20:980) and (B) formic acid/ACN/ H2O (1:950:50).

Click here for additional data file.

Supplemental Information 2 Central residue H acetylated (control group)

Motif analysis of Lung from control group. Central residue H acetylated.

Click here for additional data file.

Supplemental Information 3 Central residue H acetylated in group treated with 540 mg/Kg/day of Diacetyl

Motif analysis of Lung from group treated with 540 mg/Kg/day of Diacetyl. Central residue H acetylated.

Click here for additional data file.

Supplemental Information 4 Central residue K acetylated in group treated with 540 mg/Kg/day of Diacetyl

Motif analysis of Lung from group treated with 540 mg/Kg/day of Diacetyl. Central residue K acetylated.

Click here for additional data file.

Supplemental Information 5 Central residue K acetylated in control group

Motif analysis of Lung from control group. Central residue K acetylated.

Click here for additional data file.

Supplemental Information 6 Central residue R acetylated in group treated with 540 mg/Kg/day of Diacetyl

Motif analysis of Lung from group treated with 540 mg/Kg/day of Diacetyl. Central residue R acetylated.

Click here for additional data file.

Supplemental Information 7 Central residue R acetylated in control group

Motif analysis of Lung from control group. Central residue R acetylated.

Click here for additional data file.

Supplemental Information 8 Western blotting

Western Blotting individual data.

Click here for additional data file.

List of abbreviations

ANXA2 Annexin-A2

ANXA5 Annexin-A5

ASS1 Argininosuccinate synthase

BHMT1 Betaine-homocysteine S-methyltransferase-1

CALR Calreticulin

CHAPS 3-[(3-cholamidopropyl)dimethylammonio]-1-propanesulfonate

ES1D Carboxylesterase-1D

DTT Dithiothreitol

DNAH1 Dynein heavy chain-1

EF2K Eukaryotic elongation factor-2 kinase

HSP90B1 Endoplasmin

EPHA6 Ephrin type-A receptor

FABPL Fatty acid-binding protein

GAPDH Glyceraldehyde-3-phosphate dehydrogenase

MGEA5 O-GlcNAcase

PARK7 Deglycase DJ-1

PARS2 Protein Probable proline-tRNA ligase PBS] Phosphate-buffered saline

PMSF Phenylmethylsulfonyl fluoride

SIPA1L1 Signal-induced proliferation-associated 1-like protein-1

STIP1 Stress-induced-phosphoprotein-1

TUBB4B Tubulin beta-4B chain

TFA Trifluoroacetic acid

UBR4 Ubiquitin-protein ligase

UGT2B UDP-glucuronosyl- transferase-2B2.

Additional Information and Declarations

Competing Interests

Author Contributions

Animal Ethics

Data Availability

The authors declare there are no competing interests

Leticia Dias Lima Jedlicka conceived and designed the experiments, performed the experiments, analyzed the data, prepared figures and/or tables, authored or reviewed drafts of the paper, approved the final draft.

Sheila Barreto Guterres conceived and designed the experiments, performed the experiments, authored or reviewed drafts of the paper, approved the final draft.

Aleksandro Martins Balbino and Giuseppe Bruno Neto performed the experiments, authored or reviewed drafts of the paper, approved the final draft.

Richardt Gama Landgraf, Liliam Fernandes, Emanuel Carrilho and Etelvino José Henriques Bechara conceived and designed the experiments, analyzed the data, contributed reagents/materials/analysis tools, authored or reviewed drafts of the paper, approved the final draft.

Nilson A. Assuncao conceived and designed the experiments, analyzed the data, contributed reagents/materials/analysis tools, prepared figures and/or tables, authored or reviewed drafts of the paper.

The following information was supplied relating to ethical approvals (i.e., approving body and any reference numbers):

This study was approved by the Ethical Committee of the School Medicine of the Federal University of São Paulo.

The following information was supplied regarding data availability:

Project Name: Wistar rat, lung, NanoLC-MS/MS

Project accession: PXD004504

https://www.ebi.ac.uk/pride/archive/projects/PXD004504.

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
