# Peer review of "Increased chemical acetylation of peptides and proteins in rats after daily ingestion of diacetyl analyzed by Nano-LC-MS/MS"

_PeerJ, doi:10.7717/peerj.4688_

## Round 0.1 · original submission · Minor Revisions

As you can see, three reviewers gave very positive comments, but raised some minor concerns. Please address all critiques and revise manuscript accordingly.

Reviewer 1 ·

Basic reporting

Some parts need to be edited:

Page 11: Paragraph starting at Line 239 and ending at Line 244 -
Needs to be reframed as it is not immediately apparent to the reader what the authors are exactly trying to convey and how they are interpreting the results mentioned in this paragraph.

Page 11: Paragraph starting at Line 245 and ending at Line 253 -
This entire paragraph needs to be re-written as there are multiple instances of typographic and grammatical errors making it difficult to understand.

Page 11: Line 254 - "post-transductional" should be post-translational


Page 17: Line 388 to Line 390 is same as Line385 to Line387 - should be corrected appropriately.

Experimental design

The authors treated mice with 540 mg/Kg/day of diacetyl for 4 weeks to check the effect of this chemical on lung proteins. This amount used is substantial and difficult to imagine that anything near to this quantity is inhaled by workers who work in diacetyl manufacturing (as mentioned in the text).

The authors need to mention how they arrived at this quantity of the chemical and the length of time (4 weeks) of administration. Have they used different doses and/or length of time of administration? If so, then what is the cut-off (for both quantity and time) to observe a significant change in the acetylation levels of lung-proteins.

Validity of the findings

No comment

Reviewer 2 ·

Basic reporting

For the most part the paper is well written. There is a need for editing by a English native speaker. The paper cites pertinent literature and the results are consistent with the hypothesis that, in animal, diacetyl breakdown leads to radical formation as evidenced by the acetylation of histidinyl and arginyl side chains.

Figure 2. The figure I received for review has poor resolution. More importantly, it is difficult for the reader to figure out what is inside the purple ovals because the authors chose black as the color of the font. Please modify the figure to make sure that everything in it is readable.
Figure 3. The authors could use different colors instead of different hues of the same color. That would make it so much simpler to read.
Figure 4. Delete this figure; the information in it could be readily incorporated into the text. Should the authors argue for keeping the figure, please modify the y axis of part A by substituting 'Molecular Mass (kDa)' for the current 'Protein amount KDa'
The authors share the experimental data as supplemental information

Experimental design

The hypothesis that radical acetylation of proteins occurs in cells fed acetyl radical sources such as diacetyl was validated. The experimental design is straightforward and the data support the conclusion that diacetyl catabolism increases the level of acetylated proteins.

Validity of the findings

No comment

Additional comments

The idea of potentially expanding protein acetylation to include Arg and His side chains is of interest if it were mediated by enzymes. This paper should encourage investigators in this area of research to contrive experiments to test this possibility.

Reviewer 3 ·

Basic reporting

The article is clear with professional english used throughout and succinct.Data described supports hypothesis and is presented in unambiguous manner.

Experimental design

The experimental design is pretty well throughout and relevant network analysis is performed to study classes of proteins. Methods used for nano-LC MS/MS analysis, data collection and deconvolution are described clearly.
The authors should provide a explanation for using a very wide mass tolerance range: upto 1 Dalton and if that would have any impact of significance described between control and acetylated peptides.

Validity of the findings

no comments

---

## Round 0.2 · accepted · Accept

Thank you for addressing all concerns of the reviewers and for corresponding revision.

#